# Turning Point of Cognitive Decline for Chinese Older Adults from a Longitudinal Analysis: Protective Factors and Risk Factors

**DOI:** 10.3390/healthcare10112304

**Published:** 2022-11-17

**Authors:** Guangming Li, Kunmei Li

**Affiliations:** 1Key Laboratory of Brain, Cognition and Education Sciences, Ministry of Education, South China Normal University, Guangzhou 510631, China; 2School of Psychology, Center for Studies of Psychological Application, Guangdong Key Laboratory of Mental Health and Cognitive Science, South China Normal University, Guangzhou 510631, China; 3School of Information, Guangdong Communication Polytechnic, Guangzhou 510650, China

**Keywords:** Chinese older adults, cognitive function, cognitive decline, development trajectory, cohort-sequential design

## Abstract

Objectives: To explore the turning point of cognitive decline in Chinese older adults and to explore the influencing factors including covariates. Participants: Aged 65 and older whose cognitive function was normal at their first test. Methods: a secondary analysis that identified participants from the database of the Chinese Longitudinal Healthy Longevity Survey (CLHLS). Cohort-sequential design was used to categorize the data by age (rather than study wave), including the follow-up data of Chinese older adults aged 65–79 years and spanning 14 years. Cognitive function in 1278 participants was assessed using the Chinese Mini-Mental State Examination (CMMSE) in five waves over 14 years. Piecewise latent growth curve modeling was used to analyze the data. Results: (1) The turning point of cognitive decline in Chinese older adults occurs between the ages of 68 and 70. (2) There are statistically significant individual differences in the initial level of cognitive function and the growth rate of cognitive function before and after the transition stage. (3) Factors influencing cognitive function include residence, education level, smoking, drinking, exercise, leisure activities, social activities, Activities of Daily Living (ADL), and Instrumental Activities for Daily Living (IADL). (4) Exercise and ADL are the main protective factors, while smoking and drinking are the main risk factors. Conclusions: There is a transition stage (68–70) in the decline of cognitive function in Chinese older adults and four main factors (such as smoking, drinking, exercise and ADL) have impacts on the cognitive decline. We should strengthen these protective factors (exercise and ADL) for the cognitive decline of older adults and avoid these risk factors (smoking, drinking). To prevent the decline of the cognitive function of older adults, the government should build more places conducive to activities for older adults and actively encourage older adults to improve their physical activity level. Given our findings, public health interventions centered on alcohol and tobacco cessation in older adults should be governmentally endorsed.

## 1. Introduction

At present, more than 70 countries have entered an aging society, and China is one of them. According to the statistical bulletin of national economic and social development in China, by the end of 2019, there were 254 million older adults over the age of 60, accounting for 18.1% of the Chinese total population. It is estimated that by 2033, the scale of older adults in China will reach 400 million (Jia et al., 2020) [1]. With the continuous increase in the global older adults, a series of public health and psychological problems, which are related to the cognitive function of older adults, seriously affect the physical and mental health and quality of life of older adults (Han et al., 2022) [2]. Some problems have gradually become prominent, such as Alzheimer’s disease. Alzheimer’s disease not only severely affects the physical and mental health of older adults and their life quality in their later years but also causes a heavy socio-economic burden on the family and society.

Cognitive function, including perception, attention, memory, thinking and other abilities, is the psychological function of the human brain to receive external information, and recognize and reflect objective things (An & Liu, 2016) [3]. The cognitive function of older adults is one of the key indicators to measure their physical and mental health, which is closely related to their quality of life. Neuroscience research studies show that the cognitive function of older adults gradually declines with the age (Zaninotto et al., 2018) [4]. Cognitive decline is not necessarily a disease. Rather, it can be an early manifestation of dementia. If the cognitive decline is severe enough, it will eventually develop into Alzheimer’s disease or other diseases. At present, there is no clear and effective radical cure for Alzheimer’s disease in the medical community. However, investigating initial cases of mild to moderate disorder can stop the steady decline in cognitive processes. Therefore, it is of great significance to improve the cognitive function of older adults to prevent the occurrence and development of major neurocognitive disorders (e.g., Alzheimer’s disease). Cognitive decline in older adults can be regarded as pre-symptoms of serious health issues, a reversible transition state from normal cognitive function to Alzheimer’s. Older adults with cognitive impairment may improve their cognitive functions but may also suffer from senile dementia. Several studies have shown that prevention and early intervention in mild cases can improve older adults’ cognition (and, obviously, other associated domains) (Li & Chen, 2003; Drag & Bieliauskas, 2010; Wu et al., 2016) [5,6,7]. Thus, detecting cognitive decline in older adults is the key to preventing cognitive dysfunction or impairment in older adults.

Through longitudinal research, different scholars have studied different models and concluded that with the increase in age, the cognitive function trajectory of older adults declines at different speeds. Jonkman et al. (2018) [8] use a polynomial regression model to find that the cognitive function of older adults showed a downward trend with age. Michael et al. (2017) [9] use the latent growth curve model to find that the higher the age, the faster the cognitive function of older adults declines. The results of Zaninotto et al. (2018) [4] show that there is an inconsistent trend in the decline rate of the overall cognitive function of older adults around the age of 70–80.

Although some longitudinal studies have shown that there is a turning point of cognitive decline in older adults (Petersen et al., 2001; Zaninotto et al., 2018) [4,10], this turning point has not been clearly explored, and it is still unknown which specific age is the real turning point. Therefore, it is necessary for us to understand this turning age and its influencing factors for the cognitive decline of older adults, because it is very important and significant to recognize the turning point, which is conducive to being prepared in advance for the cognitive decline of older adults. Preventing or delaying the early onset of dementia will reduce its burdens.

However, most of the previous studies were based on cross-sectional studies (Li & Chen, 2003; Drag & Bieliauskas, 2010; Wu et al., 2016) [5,6,7]. Even longitudinal studies rarely discussed the specific time turning point of cognitive decline in the elderly, and rarely analyzed its influencing factors (Han et al., 2022; Hou et al., 2018) [2,11], which is insufficient. Therefore, it is meaningful to study the turning point of cognitive decline in Chinese older adults and its main influencing factors, which is conducive to preparing Chinese older adults and their service personnel in advance.

The following hypotheses are proposed:

**Hypothesis** **1.**
*Investigate the general trajectory of the cognitive function development of older adults, and on this basis, examine the turning point of the cognitive decline of older adults.*


**Hypothesis** **2.**
*Including time-invariant covariates and time-varying covariates to investigate their influencing factors (such as protective factors and risk factors) on the cognitive decline of older adults.*


The purpose of this paper is as follows: (1) to find out the turning point of cognitive decline in Chinese older adults by cohort-sequential design through longitudinal research; (2) to find out the risk factors and protective factors that affect the decline of cognitive function in Chinese older adults through the Piecewise Latent Growth Curve Model (PLGCM).

## 2. Methods

### 2.1. Design

A longitudinal study design has many advantages (Liu & Zhang, 2005) [12], and the most prominent one is that it can examine the trend of a certain characteristic of the same individual over time, which is to analyze the development trajectories of its behavior within the individual. However, in practical applications, a longitudinal study for a single group will encounter various problems, such as time constraints, the loss of subjects, practice effects, and the cost of investigations. Therefore, cohort-sequential design has been applied to address these shortcomings. Cohort-sequential design, also known as accelerated design, is a design method that combines cross-sectional research design and follow-up research design (Duncan et al., 1999; Liu & Zhang, 2005; Ray, 2018) [12,13]. Cohort-sequential design of Duncan et al. (1999) [13] is displayed in Table 1. That is, independent age groups have limited repeated measurements. 

The cohort-sequential design includes three age groups from 12 to 16 years old (the initial measurement age is 12, 13, and 14 years, respectively). As can be seen from Table 1, the total measurement includes information from 12 to 16 years old. In the cohort-sequential design, the age group should be selected so that the measurement time and test age are “approximately staggered”, which means that the average age of the first age group at the second measurement is approximately equal to the average age of the second age group at the first measurement, the average age of the first age group at the third measurement is approximately equal to the average age of the second age group at the third measurement and the third age group at the first measurement, and so on.

In the cohort-sequential design, different groups are interlaced in age, so it can be approximately regarded as a quasi-tracking research design that includes the age span of 12 to 16 years. In addition, data of different age groups represent different types of “missing” data. For example, for the 12-year-old group, there is no data of 15 and 16 years old; for the 13-year-old group, there is no data of 12 and 16 years old; for the 14-year-old group, there is no data for 12 and 13 years old. The data of different age groups provide information for different parts of the overall development model. To construct a complete development curve model, it is necessary to use the information of multiple groups at the same time. According to the research results of Liu & Zhang (2005) [12], the complete curve constructed by the cohort-sequential design is close to the curve obtained by the single long-term tracking study, which means that there is no significant difference between the groups in the cohort-sequential design and the real tracking study design. This shows that the cohort sequential design is effective.

The PLGCM is an extension of the Latent Growth Curve Model, which allows the combination of separate growth curves corresponding to multiple developmental stages, and repeated observations can be made from these stages (Chou et al., 2004) [14]. The latent variable in the model is measured by multiple indicators in each measurement, and it can be defined as three latent variables, including an intercept factor and two slope factors. They each have a mean and variance. The mean value of the intercept factor denotes the average initial state, and the variance of the intercept factor indicates the degree of difference between individuals at a specific time point. The larger the value, the more obvious the initial difference between individuals. The mean value of the slope factor indicates the average growth rate among time points, and the variance of the slope factor indicates the size of the difference in the growth rate among individuals. The larger the variance, the more obvious the difference in the development trajectory between individuals. The two slope factors are different, indicating that the development speed of the previous stage is inconsistent with the speed of the latter stage.

Using the PLGCM combined with a cohort-sequential design, this study intends to explore the staged changes in the cognitive function of Chinese older adults and obtain the turning point of the cognitive function of older adults. So that those people who serve older adults can intervene in the cognitive function of older adults before the turning point in the future, which is conducive to providing older adults with a better life in their later years. On the one hand, this study aims to examine the development trajectory of cognitive function in older adults and to understand the regularity of cognitive function in older adults. On the other hand, it includes different covariates to explore the influencing factors of cognitive function, in order to improve the recognition of older adults.

### 2.2. Participants

This study examined a sample of Chinese older adults who were followed over 14 years. Data were collected in five waves to examine the developmental trajectories of the cognitive function of older adults. We also considered a number of risk and protective factors (e.g., substance use and leisure activities) that might affect the cognitive function of older adults. In this study, the longitudinal study method, combined with the cohort-sequence design (Ray, 2018) [15], was used to analyze the data from a healthy longevity survey about older adults with a span of about 14 years. This study was divided into two sub-studies. In the first study, PLGCM was constructed to explore the cognitive function of older adults and the general developmental trajectories and transition stages of each dimension. In the second study, time-varying covariates and non-time-varying covariates were included to explore the impacts of these covariates on the cognitive function development of Chinese older adults.

Data were drawn from the Chinese longitudinal healthy longevity survey (CLHLS). Primary modules in the CLHLS included personal and family background, social activities, mental status, and the source of income. The survey covered 631 county-level administrative districts in 23 provinces and municipalities across the country. These provinces covered 85% of the country’s total population (Zeng, 2008) [16]. The data used in this study were collected in five waves: 2005, 2008, 2011, 2014, and 2018. Like other studies (Ray, 2018) [15], the data were processed using a cohort-sequential design (Liu & Zhang, 2005) [12] and the current study sample consisted of three cohorts defined by age at baseline including 65–67-year-old (*n* = 1278). The first cohort selected subjects who had completed three, four, or five tests between 2005 and 2018, with a baseline age of 65–67. The second cohort selected subjects who had completed three or four tests between 2008 and 2018, with a baseline age of 68–70. The third cohort selected subjects who completed three tests between 2011 and 2018, with a baseline age of 71–73. A total of 714 participants (55.9%) contributed data over all 5 age groups, and the missing data in each group are displayed in Table 2.

Characteristics of the study participants among 1278 older adults in 5 waves are shown in Table 3.

To describe the basic information of data, the descriptive statistics are shown in Table 4.

To indicate whether the data is randomly missing or not, we performed cross-tab chi-square tests on the selected data, and the chi-square value was not significant, χ^2^(df) = 36.708, *p* = 0.436, indicating that the missing data were missing at random (Little, 1988) [17]. This study included a sample of 1278 older adults for analysis which was divided into five groups based on ages, ranging from 65 to 79 years old, including 65–67 years old, 68–70 years old, 71–73 years old, 74–76 years old and 77–79 years old.

### 2.3. Measures

#### 2.3.1. Cognitive Function

Cognitive function was assessed in the CLHLS using the Mini Mental State Examination Scale (MMSE) (Folstein et al., 1975) [18]. The 24-item scale was translated and modified into a Chinese version to adapt to the Chinese cultural background. The Chinese version had been demonstrated to have satisfactory reliability and validity (e.g., Du et al., 2019; Ni et al., 2020; Yu et al., 2021; Zhong et al., 2017) [19,20,21,22]. Five domains of cognitive function were measured: general ability (e.g., ‘what month is it now’); reaction capacity (e.g., ‘repeat the three objects in order after me quickly’); attention and computation (e.g., simple mathematical problems); memory (e.g., ‘repeat the three objects just learned before in order again’); and Language comprehension and coordination (e.g., ‘write a complete and meaningful sentence including subject and verb’). Items were scored as 1 if the answers were correct, and 0 for an incorrect or ‘unable to’ answer, except for the sixth question (full score is 7). Accordingly, the CLHLS cognitive function score ranged from 0 (all answers are incorrect or ‘unable to’) to 30 (all answers are correct). The more correct answers an individual made, the better his/her cognitive function was. Cronbach’s alpha coefficients ranged from 0.78 to 0.90 through the five waves data.

#### 2.3.2. Covariates

Covariates included: residence (1 = city; 2 = town; 3 = township), gender (male = 0; female = 1), living style (living with others = 1, otherwise 0), education level (years of education), chronic diseases (with any of 26 chronic diseases including hypertension, diabetes, stroke, cerebrovascular disease, et al.; any disease = 1; no disease = 0), smoking (frequent smoking = 1; no smoking = 0), drinking (frequent drinking = 1; no drinking = 0), exercise (frequent exercise = 1; no exercise = 0), physical labor (frequent labor = 1; no labor = 0), leisure activities (the higher the score, the worse the leisure activities), social activities (social activities with others = 1; no social activities = 0), Activities of Daily Living (ADL) (e.g., do you need help in the shower; higher scores indicating worse activity ability), and Instrumental Activities for Daily Living (IADL) (e.g., can you travel by public transport alone; higher scores indicating worse activity ability).

### 2.4. Analysis Procedure

We adopted a cohort-sequential design (Liu & Zhang, 2005; Orth, Trzesniewski, & Robins, 2010) [12,23] and categorized the data based on age (Shown in Table 1). Descriptive statistics and inferential statistical analyses were performed in SPSS 28.0. Primary analysis models were conducted in Mplus 8.0 (Muthén & Muthén, 2017) [24] using maximum likelihood estimation with robust standard errors. The following indices were used to examine model fit: χ²/df, compare the fit index (CFI; good > 0.95, acceptable > 0.90) and the Tucker–Lewis index (TLI; good > 0.95, acceptable > 0.90), the root mean square approximate error (RMSEA; good < 0.06, acceptable < 0.08), and the root mean square residual (SRMR; good < 0.05, acceptable < 0.08) (Browne & Cudeck 1992; Hu & Bentler 1999) [25,26].

## 3. Results

### 3.1. The Development Trajectory of Cognitive Function in Older Adults

We first conducted the PLGCM to estimate the development trajectory of cognitive function in Chinese older adults. According to Karevold et al. (2012) [27], the PLGCM generates three potential factors: an intercept factor and two slope factors. The two slope factors in our model were expressed as linear growth in both the early stage (slope 1: from 65 to 70 years old) and the later stage (slope 2: from 71 to 79 years old). Therefore, the endpoint of slope 1 and the start point of slope 2 (68–70 years old) were set as the age range. In order to illustrate the characteristics of PLGCM, we analyzed the fitting indicators of the model. In the PLGCM, the model fit indices indicated a good fit to the data (χ² = 32.06, df = 7, χ²/df = 4.6; CFI = 0.94; TLI = 0.90, RMSEA = 0.06, and SRMR = 0.04). As shown in Table 5, cognitive function showed a gentle downward trend in the early stage (M = −0.392, SE = 0.182, *p* = 0.045) but a rapid downward trend in the later stage (M = −0.509, SE = 0.069, *p* = 0.0008). The variance of the cognitive function intercept in the model was 1.641 (SE = 0.603, *p* = 0.009), and the variances of slop 1 and slope 2 were 4.096 (SE = 2.090, *p*= 0.048) and 0.637 (SE = 0.312, *p* = 0.047), respectively, indicating obvious individual differences in the initial level and growth rate of cognitive function in older adults. The model diagram is shown in Figure 1.

The downward trend chart of cognitive function and its five different dimensions including general ability, reaction capacity, attention and computation, memory, language comprehension and coordination are shown in Figure 2.

### 3.2. Conditional PLGCM of Cognitive Function in Older Adults

We then considered the effects of a number of time-invariant and time-variant variables on the intercepts and slopes of the PLGCM of cognitive function in Chinese older adults. Conditional PLGCM of cognitive function variables are shown in Table 6 (time invariant covariates) and Table 7 (time-varying covariates).

As shown in Table 6, living in a city (*β* = −0.496, SE = 0.131, *p* < 0.001) and having a high level of education (*β* = 0.089, SE = 0.020, *p* < 0.001) are associated with a higher baseline level of cognitive function. With aging, the cognitive function level of older adults shows a downward trend and decreases rapidly after the age of 68–70. Education level significantly predicted (*β* = 0.032, SE = 0.016, *p* < 0.01) the change rate of cognitive function in older adults after the age of 68–70 (see Table 6). A high level of education (β = 0.032, SE = 0.016) has a significant positive predictive effect on the change rate of cognitive function level after the turning point. That is, the cognitive function level of Chinese older adults with high education level declines more rapidly. Gender, residence style and chronic diseases were not related to the baseline level and rate of change of cognitive function.

As shown in Table 7, smoking, drinking, exercise, leisure activities, social activities, ADL and IADL all have impacts on the cognitive function of older adults. At the age of 68–70 and 74–76, the cognitive function of older adults is affected by smoking (*β* = −1.534, SE = 0.537, *p* < 0.01 and *β* = 0.984, SE = 0.375, *p* < 0.05, respectively) and exercise (*β =* −1.382, SE = 0.468, *p* < 0.01; *β* = 1.037; SE = 0.353, *p* < 0.01, respectively). Drinking significantly affects the cognitive function of older adults aged 68–70 (*β* = −1.293, SE = 0.598, *p* < 0.05). Similarly, leisure activities have a significant impact on the cognitive function of older adults after the age of 70. Social activities significantly affect the cognitive function of older adults aged 74–76 (*β* = 0.301, SE = 0.126, *p* < 0.05). ADL can predict the cognitive function of older adults after the age of 68. No matter what age, IADL can affect the cognitive function of older adults (the absolute regression coefficient is between 0.3~0.6).

## 4. Discussion

### 4.1. The Development Trajectory of the Cognitive Function of Chinese Older Adults

This study observed that the development trajectory of the cognitive function of older adults showed a slow downward trend over time before the transition stage (aged 68–70), and a significant and rapid decline after it. There are significant individual differences in the initial level and the growth rate of cognitive function before and after the transition stage.

Before this stage, the cognitive function of older adults shows a downward trend and it obviously accelerates after the age of 68–70, which is consistent with Finkel et al. (2003) [28] who found through simulation research that the transition stage of rapid cognitive decline in older adults is around 68–70 years old. The rapid cognitive decline may be caused by disease or degradation of physical function. Another possible reason may be that around the ages of 68–70, the mental state of older adults undergoes some life changes that happen and contribute to lower mental stimulation (e.g., retirement, change of highly stimulating professions to more relaxed environments/activities), and then develop a sense of aging (Chen et al., 2018; Ni et al., 2020) [20,29]. Some factors, such as slowness in action, slower reactions, the whitening of hair, the loss of teeth, or the special treatment of people around them and verbally calling them old people, etc., can force them to accept life changes (e.g., sometimes they feel that they are old). As a result, their cognitive function after the age of 68–70 decreases significantly.

In terms of individual differences in the initial level and change in speed of cognition function, this study reports that the changes in the cognitive function of older adults varied from person to person over time. Older adults show differences in many aspects, such as education level, leisure style in their later life, growth experience, and so on. All these may be the reasons for the individual differences in the initial level and development trajectories of cognitive function in older adults.

### 4.2. Factors Influencing the Development Trajectory of Cognitive Function

Living in the city is one of the protective factors for the cognitive function decline of older adults, as the cognitive function of older adults in villages and towns is worse than those in cities. The reason for this result may be that urban areas have better economic, environmental, and medical conditions, and urban older adults have better access to relevant information and attention than those in rural areas (Yang et al., 2016) [30]. City settings have a richer offer to citizens in terms of stimulating environments (e.g., senior universities, theaters, museums, libraries, etc.). Thus, it is easier for older adults in these settings to continuously engage in activities that are cognitively challenging. Hou et al. (2018) [11] proposed that the level of education would affect the cognitive function of older adults, which means that the higher the level of education, the higher the level of cognitive function. This may be because people with a high level of education can continuously stimulate the increase in the number of synapses in the cerebral cortex due to their learning and paying attention to things (Andel et al., 2005) [31], thereby reducing the risk of cognitive decline.

This study finds that older adults with low ADL and IADL also have low cognitive function. The impairment of the ability of daily activities of older adults will first lead to the reduction in their participation in social communication, which is an important way for older adults to obtain information and social support and plays an important role in physical and mental health and cognitive function of older adults. Consequently, the cognitive function of older adults will decline (Li et al., 2017) [32].

Our findings show that before and after the turning point, smoking is an influencing factor, which is consistent with some studies (Liu et al., 2002) [33]. However, in this paper, the survival age of regular smokers has not been considered. The survival age of regular smokers may be smaller than that of less frequent smokers; therefore, some regular smokers lose their cognitive ability but their data are not recorded. The cognitive function level of existing smokers is higher than that of infrequent smokers. While our findings are focused on the direct impact on older adults’ cognition, smoking at advanced ages is known to impact other health-related conditions such as lung disease, asthma, stroke, etc. (Aune et al., 2016; Larsson et al., 2020; Stefanidou et al., 2022) [34,35,36].

Exercise can significantly predict the cognitive function of older adults. Before the transition stage, older adults who exercise regularly have worse cognitive functions; however, they have better cognitive functions after it. A possible reason is that the brain activity pattern of older adults during working memory tasks is from the synchronization of low-frequency alpha events in the parietal occidental and frontal lobe to the increase in desynchronization of high-frequency alpha in the left hemisphere, so as to improve the working memory ability of older adults (Ding et al., 2020) [37].

Leisure activities are a protective factor for cognitive function. This is consistent with the research of Ni et al. (2020) [20]. The higher the participation in leisure activities, the better the cognitive function is. Participating in leisure activities is an interactive process conducive to the maintenance of social relations. After the interactive stimulation of brain cells, the brain would be in an active and excited state, which promotes the interaction ability among people. Moreover, it can also play a positive role in the body and mind to strengthen the brain, ultimately maintaining a good level of cognitive function.

Drinking is one of the risk factors for the decline of cognitive function in older adults. The cognitive function of older adults who often drink is low before the transition stage. The impact of drinking on body health is related to the amount of alcohol consumption. A small amount of drinking is conducive to body health, but excessive drinking will harm body health. This study did not take into account the amount of alcohol consumption (Franke et al., 2014) [38]. Future studies may consider whether individual alcohol consumption would affect cognitive function.

This study observes that older adults with regular participation in social activities have a higher score on cognitive function, which is consistent with the research of Han et al. (2022) [2]. Regular participation in social activities is beneficial to maintaining cognitive function. A possible reason for this is that regular social interaction is a good way to maintain frequent and positive interactions, conducive to a lasting interpersonal relationship. Ultimately, this helps older adults maintain a high level of cognition function. Therefore, families and communities can help older adults establish inter-generational relationships and social networks and provide strong spiritual support for older adults.

We can distinguish between risk factors and protective factors. Risk factors include smoking, and drinking and protective factors include residence, education level, exercise, leisure activities, social activities, ADL and IADL. According to Table 5 and Table 6, *p* values of residence, education level, smoking, drinking, exercise, leisure activities, social activities, ADL and IADL are significant, but if we consider the absolute *β* ≥1.00, then the main risk factors are smoking (C_2: *β* = −1.534, SE = 0.537) and drinking (C_2: β = −1.293, SE = 0.598); the main protective factors are exercise(C_2: *β =* −1.382, SE = 0.468) and ADL (C_5: β = −1.326, SE = 0.193). Therefore, we should try our best to let older adults avoid smoking and drinking and strength their exercise and ADL. So, to prevent the decline of the cognitive function of older adults, respect from the government should be reflected in the accessibility of public buildings and spaces and in the range of opportunities that the city offers to older adults for social participation, entertainment, volunteering, etc. Given our findings, public health interventions centered on alcohol and tobacco cessation in older adults should be governmentally-endorsed. People should encourage older adults to participate in more social activities and persuade them to smoke and drink less.

Using the PLGCM and cohort-sequential design that are new and creative, the turning point of cognitive decline in Chinese older adults is found, which has not been studied before. We find the two main risk factors and the two main protective factors influencing cognitive function in Chinese older adults, which is different from other studies.

Compared with previous studies (Han et al., 2022; Hou et al., 2018; Petersen et al., 2001; Zaninotto et al., 2018) [2,4,10,11], the contribution of this study is mainly shown in the following two points: (1) The specific turning point of cognitive decline of the elderly was found, namely 68–70. In fact, understanding the turning point of the trajectory is of great significance to the workers for older adults and older adults themselves; (2) Four main factors influencing the turning point of cognitive decline in the elderly were explored. Before the turning point, protective interventions on cognitive function can protect against the decline of cognitive function.

This research explores the development of the cognitive function of older adults and can help us understand the development model of the cognitive function of older adults, and help clarify the work direction for older adults, and provide the theoretical basis for improving the comfort of older adults’ lives.

### 4.3. Limitations

First of all, we considered several main covariate variables, but other variables could also be included. Secondly, the data were elders self-reported and self-evaluated. Future studies should include objective measures to assess self-rated health. Thirdly, this study did not consider whether the surviving age of frequent smokers will affect cognitive function. Future research is warranted to examine the impact of surviving age on the cognitive function of smoking older adults. The results of smoking around 68–70 should be considered their study limitations (e.g., study design, covariate definition, interpretation bias, and data collection). Finally, in this study, we used piecewise growth modeling. When we entered several covariates, individual differences existed. Future studies could use piecewise growth mixture modeling to classify participants into different categories (membership) and then explore the study relationships.

## 5. Conclusions

(1)There is a transition stage in the decline of cognitive function in older adults. The cognitive function of older adults showed a slow downward trend before the age of 68–70, and a rapid downward trend after the age of 68–70.(2)There are significant individual differences in the intercept and slope of the cognitive function of older adults. There are significant individual differences in the initial level of overall cognitive function and the growth rate before and after the transition period.(3)Several covariates can impact the development trajectories of cognitive function. The influencing factors of cognitive function include residence, education level, smoking, drinking, exercise, leisure activities, social activities, ADL, and IADL.(4)Exercise and ADL are the main protective factors but smoking and drinking are the main risk factors. Through the longitudinal analysis, we find the main risk factors and the main protective factors. We should strengthen these protective factors for older adults and avoid these risk factors.

## Figures and Tables

**Figure 1 healthcare-10-02304-f001:**
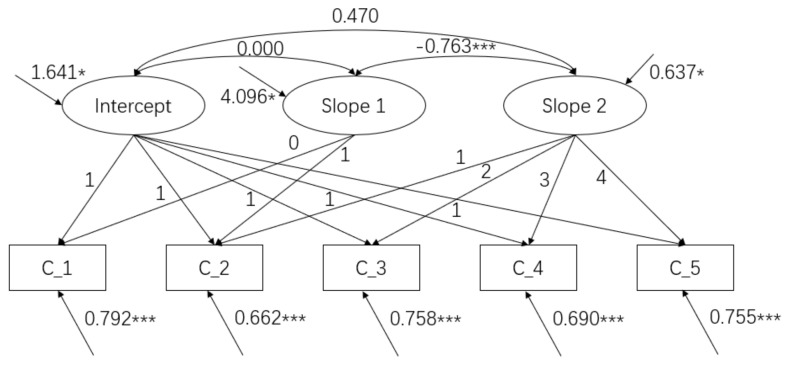
PLGCM diagram of cognitive function. Note: C_1 = 65−67 years old; C_2 = 68−70 years old; C_3 = 71−73 years old; C_4 = 74−76 years old; C_5 = 77−79 years old; * *p* < 0.05; *** *p* < 0.001.

**Figure 2 healthcare-10-02304-f002:**
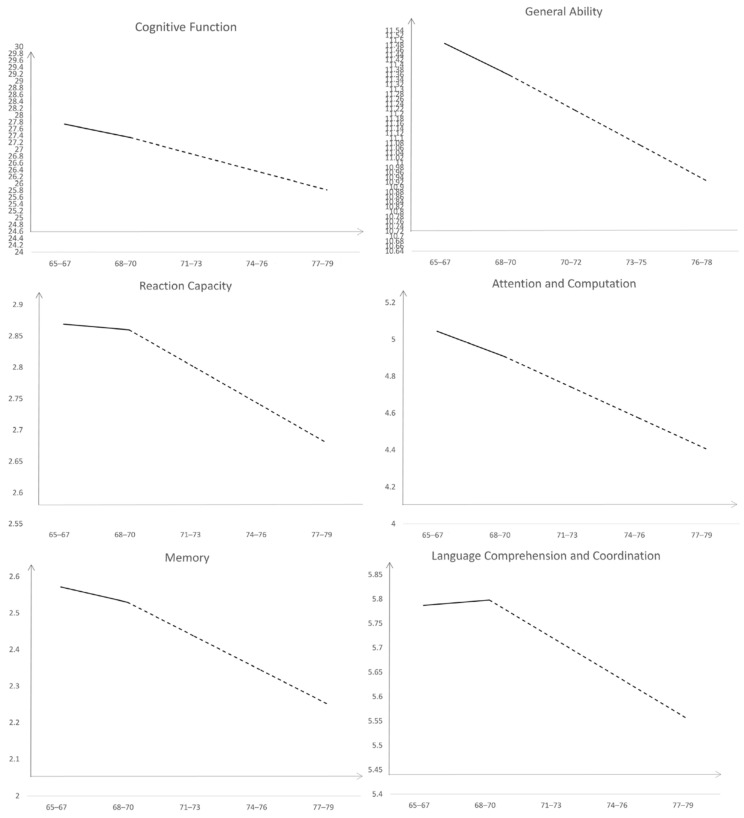
Downward trend charts of cognitive function and five different dimensions.

**Table 1 healthcare-10-02304-t001:** An example of cohort-sequential design.

Group	Age at the Time of Test (Years Old)				
	12 Years Old	13 Years Old	14 Years Old	15 Years Old	16 Years Old
12 years old	√	√	√		
13 years old		√	√	√	
14 years old			√	√	√

**Table 2 healthcare-10-02304-t002:** Sample size distribution (per cohort) during the study period.

Generation	65–67 Years Old	68–70 Years Old	71–73 Years Old	74–76 Years Old	77–79 Years Old
65–67 years old	743	419	419	369	248
68–70 years old	/	552	552	448	218
71–73 years old	/	/	303	294	290
Valid data	743	971	1274	1111	756
Missing data	535	307	4	167	522
Total	1278	1278	1278	1278	1278

**Table 3 healthcare-10-02304-t003:** Characteristics of the study participants among 1278 older adults in 5 waves.

Variables	Attribution	*n* (%)
Gender	male	537 (42.0%)
	female	741 (58.0%)
Resident	city	194 (15.2%)
	town	298 (23.3%)
	country	786 (61.5%)
Way of living	with household member(s)	1062 (83.1%)
	alone	211 (16.5%)
	in an institution	5 (0.4%)
Chronic Diseases	ill	8.8%
	Not sick	91.2%
Educational Level	0 year	650 (50.8%)
	1 year	53 (4.1%)
	2 year	90 (7.0%)
	3 year	79 (6.2%)
	4 year	66 (5.2%)
	5 year	67 (5.2%)
	6 year	130 (10.2%)
	7 year	17 (1.3%)
	8 year	33 (2.6%)
	9 year	37 (2.6%)
	10 year	17 (1.3%)
	11 year	3 (0.2%)
	12 year	19 (1.5%)
	13 year	4 (0.3%)
	14 year	1 (0.1%)
	15 year	4 (0.3%)
	16 year	4 (0.3%)
	17 year and above	4 (0.3%)

**Table 4 healthcare-10-02304-t004:** Descriptive statistics.

	N	Mean	SD	C_1	C_2	C_3	C_4	C_5
C_1	743	27.89	2.813	1				
C_2	971	26.75	3.803	0.208 **	1			
C_3	1273	26.43	4.665	0.231 **	0.263 **	1		
C_4	1111	26.29	4.727	0.187 **	0.261 **	0.267 **	1	
C_5	756	24.24	6.103	0.243 **	0.125 **	0.078	0.242 **	1

Note: C_1 = 65–67 years old; C_2 = 68–70 years old; C_3 = 71–73 years old; C_4 = 74–76 years old; C_5 = 77–79 years old; ** *p* < 0.01.

**Table 5 healthcare-10-02304-t005:** PLGCM of cognitive function.

**Cognitive Function**	**M Intercept**	**M Slope 1**	**M Slope 2**	**ΔIntercept**	**ΔSlope 1**	**ΔSlope 2**
	27.740 ***	−0.392 *	−0.509 ***	1.641 **	4.096 *	0.637 *

Note: * *p* < 0.05; ** *p* < 0.01; *** *p* < 0.001.

**Table 6 healthcare-10-02304-t006:** Conditional PLGCM of cognitive function variables (time-invariant covariates).

Time-Invariant Covariates	Intercept		Slope 1		Slope 2	
	*β*	SE	*β*	SE	*β*	SE
Residence	−0.496 ***	0.131	0.236	0.241	0.082	0.094
Gender	−0.334	0.203	−0.139	0.364	−0.021	0.138
Living style	−0.274	0.284	0.270	0.498	−0.047	0.187
Education level	0.089 ***	0.020	−0.051	0.039	0.032 **	0.016
Chronic diseases	−0.291	0.307	−0.210	0.578	−0.209	0.236

Note: ** *p* < 0.01; *** *p* < 0.001.

**Table 7 healthcare-10-02304-t007:** Conditional PLGCM of cognitive function variables (time-varying covariates).

Time-Varying Covariate	C_1		C_2		C_3		C_4		C_5	
	*β*	SE	*β*	SE	*β*	SE	*β*	SE	*β*	SE
Smoking	0.646	0.372	−1.534 **	0.537	0.077	0.478	0.984 *	0.375	0.375	0.829
Drinking	0.243	0.551	−1.293 *	0.598	−0.680	0.491	0.439	0.529	0.361	0.920
Exercise	0.501	0.385	−1.382 **	0.468	0.225	0.365	1.037 **	0.353	0.132	0.591
Physical labor	−0.613	0.542	−1.018	0.642	−0.136	0.434	0.537	0.394	−0.305	0.626
Leisure activities	−0.071	0.038	−0.047	0.042	−0.079 **	0.027	−0.115 ***	0.029	−0.228 ***	0.047
Social activity	0.161	0.135	−0.173	0.296	0.010	0.936	0.301 *	0.126	−0.004	0.269
ADL	1.268	2.669	0.866 *	0.430	0.201	0.453	−0.469 *	0.167	−1.326 ***	0.193
IADL	−0.323 *	0.151	−0.287 **	0.094	−0.364 ***	0.059	−0.432 ***	0.049	−0.583 ***	0.057

Note: C_1 = 65–67 years old; C_2 = 68–70 years old; C_3 = 71–73 years old; C_4 = 74–76 years old; C_5 = 77–79 years old, * *p* < 0.05; ** *p* < 0.01; *** *p* < 0.001.

## Data Availability

The data are not publicly available due to privacy restrictions.

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
