# Peer review of "Turning Point of Cognitive Decline for Chinese Older Adults from a Longitudinal Analysis: Protective Factors and Risk Factors"

_healthcare, 2022, doi:10.3390/healthcare10112304_

Round 1

Reviewer 1 Report

Dear Dr Rahman Shiri

Editor in Chief of the Healthcare Journal,

I appreciate the opportunity to review the manuscript titled “Turning Point of Cognitive Decline for Chinese Elderly from a Longitudinal Analysis: Protective Factors and Risky Factors”. The authors presented a retrospective secondary analysis that aimed to “explore the turning point of cognitive decline in Chinese elderly, and to explore the influencing factors including covariates”.

Overall, I believe the manuscript is well-structured and concise, requiring English revisions. The reference style will require revision from the authors to comply with MDPI’s format.

The manuscript’s aim fits the journal’s scope. Below I provide a few comments/questions for the authors’ analysis.

[Title, abstract and keywords]

- The last considerations made by the authors are reductive, repeating what is known from the results [c.f. “We should strengthen these protective factors (exercise and ADL) for the cognitive decline of the elderly to avoid these risky factors (smoking, drinking)”]. I believe a more proactive approach would be better, for example, highlighting the contributions of your findings to health policy planning in the country.

- I would refrain from using the term “elderly”, as this is not well-received in international publications in gerontology/geriatrics. The most common term is “older adults”. Please revise accordingly across the manuscript, including the keywords.

- I would suggest removing the keyword “Piecewise latent growth curve modelling”. I find it hard to believe that other researchers in the field use such terms in database searches.

[Introduction]

-  Line 33 “to a series of intelligent processing processes”, considering revising this sentence.

- Lines 35-39: The sentences are slightly confusing. I would advise the authors to refer to mild, moderate, and major neurocognitive disorders (e.g., Alzheimer’s disease), and how investing in initial cases of mild to moderate disorder can stop the steady decline in cognitive processes.

- Before introducing the study, the authors should include a few sentences that support its importance. As I said before, several studies have shown that prevention and early intervention in mild cases can improve older adults’ cognition (and, obviously, other associated domains). Presenting just a brief paragraph before mentioning your study is not appropriate.

- Lines 41 to 60 are wrongfully included as part of the Introduction. You are explaining what you did (and how) to the readers. This should be included as the initial part of the Methods section!

- Considering the last point (lines 41-60), I would suggest including a figure explaining your cohort-sequential design. It will improve the manuscript’s clarity for inexperienced readers.

[Methods]

- Line 69: the acronym CLHLS was already presented before (line 47). Please revise.

-  Table 3 is not mentioned before being presented. The authors should clarify what is being presented in Table 3.

- Concerning your covariates, some minor questions for clarification. What is considered “infrequent drinking” (e.g., less than once a week? Everyone once a month? Not a drinker?)? The same goes for exercise, there is a huge gap (supported by existing literature) between “frequent” exercise and no exercise. This is rather significant since your study findings revolve around the influence of such covariates.

- Lines 138-141: What was the reasoning behind the two slope factors? The difference in age intervals (and expected overall decline from the ageing process) seems odd. Please clarify the rationale behind your decision and consider including it in the manuscript.

[Results]

- Please consider revising the way you present your findings in some sections (e.g., “Drinking only affects the cognitive function of the elderly aged 68-70 (β = -1.293, SE = 0.598, p < 0.05)”). Using expressions such as “only affects” can be negatively interpreted by journal readers, some of whom can be casual citizens looking for information (e.g., drinking frequently does not affect my cognition if I am 74-76 years of age).  

[Discussion]

- Lines 195-201: This whole section should be reworded. While I do understand the authors’ viewpoint, it reads as slightly ageist and condescending (e.g., “they may realize that they are indeed old”). The authors should search for a few studies on the acceptance of life transitions and revise this topic. Also, the authors seem to omit other life changes that happen around the ages of 68-70 and contribute to lower mental stimulation (e.g., retirement, change of highly stimulating professions to more relaxed environments/activities). This is well-studied and should be referenced here.

- Lines 209: I would also include here that city settings have a richer offer to citizens in terms of stimulating environments (e.g., senior universities, theatres, museums, libraries, etc.). Thus, is easier for older adults in these settings to continuously engage in activities that are cognitive challenging.

- Line 226: This is another result that should be discussed carefully. The authors propose that smoking can “relax” older adults, and thus “protect” their cognitive function after the age of 70 years of age. The potential social implications of such a claim are significant, and I highly advise the authors to be critical of their results considering their study limitations (e.g., design, covariate definition, data collection). Moreover, it would be important to contrast your findings with other studies, which are lacking.

- The authors should address in detail other limitations related to the study design, the potential risk of interpretation bias, and the lack of a detailed instrument for data collection (which could explain some of the results found for covariates such as smoking).

- The discussion section lacks a paragraph or two focused on the implications of your findings to clinical practice and social policy. What and whom do you expect to inform with such results?

[Conclusions]

- Do not list your conclusions. Please revise.

Author Response

I appreciate the opportunity to review the manuscript titled “Turning Point of Cognitive Decline for Chinese Elderly from a Longitudinal Analysis: Protective Factors and Risky Factors”. The authors presented a retrospective secondary analysis that aimed to “explore the turning point of cognitive decline in Chinese elderly, and to explore the influencing factors including covariates”.

Overall, I believe the manuscript is well-structured and concise, requiring English revisions. The reference style will require revision from the authors to comply with MDPI’s format.

The manuscript’s aim fits the journal’s scope. Below I provide a few comments/questions for the authors’ analysis.

[Title, abstract and keywords]

- The last considerations made by the authors are reductive, repeating what is known from the results [c.f. “We should strengthen these protective factors (exercise and ADL) for the cognitive decline of the elderly to avoid these risky factors (smoking, drinking)”]. I believe a more proactive approach would be better, for example, highlighting the contributions of your findings to health policy planning in the country.

Answer: Thanks for your comments! To highlight the contributions of our findings to health policy planning in the country, a paragraph has been added to the abstract as follows: To prevent the decline of cognitive function of older adults, the government should build more places conducive to the activities for older adults and actively encourage older adults to improve their physical activity level. At the same time, the government should also do a good job in promoting older adults to quit smoking and drinking.

All revisions are marked in red in this revised paper.

- I would refrain from using the term “elderly”, as this is not well-received in international publications in gerontology/geriatrics. The most common term is “older adults”. Please revise accordingly across the manuscript, including the keywords.

Answer: Thanks for your comments! We have revised the elderly into older adults in this paper.

- I would suggest removing the keyword “Piecewise latent growth curve modelling”. I find it hard to believe that other researchers in the field use such terms in database searches.

 Answer: Thanks for your comments! We have removed the keyword “Piecewise latent growth curve modelling”.

[Introduction]

-  Line 33 “to a series of intelligent processing processes”, considering revising this sentence.

Answer: Thanks for your comments! We have revised them. All revisions are marked in red in this revised paper. Revised as :

Cognitive function, including perception, attention, memory, thinking and other abilities, is the psychological function of the human brain to receive external information, recognize and reflect objective things (An & Liu, 2016).

- Lines 35-39: The sentences are slightly confusing. I would advise the authors to refer to mild, moderate, and major neurocognitive disorders (e.g., Alzheimer’s disease), and how investing in initial cases of mild to moderate disorder can stop the steady decline in cognitive processes.

Answer: Thanks for your comments! We have revised them. All revisions are marked in red in the Introduction. Revised as : 

At present, there is no clear and effective radical cure for Alzheimer's disease and other diseases in the medical community. However, investigating in initial cases of mild to moderate disorder can stop the steady decline in cognitive processes. Therefore, it is of great significance to improve the cognitive function of older adults to prevent the occurrence and development of major neurocognitive disorders (e.g., Alzheimer’s disease).

- Before introducing the study, the authors should include a few sentences that support its importance. As I said before, several studies have shown that prevention and early intervention in mild cases can improve older adults’ cognition (and, obviously, other associated domains). Presenting just a brief paragraph before mentioning your study is not appropriate.

Answer: Thanks for your comments! We have revised them. All revisions are marked in red in the Introduction. Revised as :

Several studies have shown that prevention and early intervention in mild cases can improve older adults’ cognition (and, obviously, other associated domains) (Li & Chen, 2003; Drag & Bieliauskas, 2010; Wu et al., 2016). Thus, detecting cognitive decline in older adults is the key to preventing cognitive dysfunction or impairment in older adults.

- Lines 41 to 60 are wrongfully included as part of the Introduction. You are explaining what you did (and how) to the readers. This should be included as the initial part of the Methods section!

Answer: Thanks for your comments! We have revised them. This should be included as the initial part of the Methods section! And we adjusted them.

- Considering the last point (lines 41-60), I would suggest including a figure explaining your cohort-sequential design. It will improve the manuscript’s clarity for inexperienced readers.

Answer: Thanks for your comments! We add a figure explaining your cohort-sequential design. And please see the Table 1 (newly add).

[Methods]

- Line 69: the acronym CLHLS was already presented before (line 47). Please revise.

Answer: Thanks for your comments! We have revised it.

-  Table 3 is not mentioned before being presented. The authors should clarify what is being presented in Table 3.

Answer: Thanks for your comments! We have revised it. Revised as :

Characteristics of the study participants among 1,278 older adults in 5 waves are shown in Table 3.

To describe the basic information of data, the descriptive statistics are shown in Table 4.

- Concerning your covariates, some minor questions for clarification. What is considered “infrequent drinking” (e.g., less than once a week? Everyone once a month? Not a drinker?)? The same goes for exercise, there is a huge gap (supported by existing literature) between “frequent” exercise and no exercise. This is rather significant since your study findings revolve around the influence of such covariates.

Answer: Thanks for your comments! We have revised it. “infrequent drinking” has been changed to “no drinking” just as smoking, disease, physical labor and exercise. Revised as :

smoking (frequent smoking = 1; no smoking = 0), drinking (frequent drinking = 1; no drinking = 0).

- Lines 138-141: What was the reasoning behind the two slope factors? The difference in age intervals (and expected overall decline from the ageing process) seems odd. Please clarify the rationale behind your decision and consider including it in the manuscript.

Answer: Thanks for your comments! This is determined based on experience, e. g., according to Karevold et al. (2012), PLGCM generates three potential factors: an intercept factor and two slope factors. In particular, this paper only studies one turning point. In addition, some litterateurs also show that there is only one inflection point in the cognitive function of the elderly, e. g., some studies have shown that there is a turning point of cognitive decline in older adults (Petersen et al., 2001; Zaninotto et al., 2018).

[Results]

- Please consider revising the way you present your findings in some sections (e.g., “Drinking only affects the cognitive function of the elderly aged 68-70 (β = -1.293, SE = 0.598, p < 0.05)”). Using expressions such as “only affects” can be negatively interpreted by journal readers, some of whom can be casual citizens looking for information (e.g., drinking frequently does not affect my cognition if I am 74-76 years of age).  

Answer: Thanks for your comments! We have removed the “only”. It is inappropriate to use “only” here. Other similarities have also been modified.Revised as :

Drinking significantly affects the cognitive function of older adults aged 68-70 (β = -1.293, SE = 0.598, p < 0.05). Similarly, Leisure activities have a significant impact on the cognitive function of older adults after the age of 70.

[Discussion]

- Lines 195-201: This whole section should be reworded. While I do understand the authors’ viewpoint, it reads as slightly ageist and condescending (e.g., “they may realize that they are indeed old”). The authors should search for a few studies on the acceptance of life transitions and revise this topic. Also, the authors seem to omit other life changes that happen around the ages of 68-70 and contribute to lower mental stimulation (e.g., retirement, change of highly stimulating professions to more relaxed environments/activities). This is well-studied and should be referenced here.

Answer: Thanks for your comments! We have revised them as follows:

Another possible reason may be that around the ages of 68-70, the mental state of older adults undergone some life changes that happen and contribute to lower mental stimulation (e.g., retirement, change of highly stimulating professions to more relaxed environments/activities), and then develop a sense of aging (Chen et al., 2018; Ni et al., 2020). Some factors, such as slowness in action, the slower reactions, the whitening of hair, the loss of teeth, or the special treatment of people around them and verbally calling them old people etc., can cause that they have to accept life changes ( e. g., sometimes they feel that they are old). As a result, their cognitive function after the age of 68-70 decreases significantly. 

- Lines 209: I would also include here that city settings have a richer offer to citizens in terms of stimulating environments (e.g., senior universities, theatres, museums, libraries, etc.). Thus, is easier for older adults in these settings to continuously engage in activities that are cognitive challenging.

Answer: Thanks for your comments! We have revised them as follows:

City settings have a richer offer to citizens in terms of stimulating environments (e.g., senior universities, theaters, museums, libraries, etc.). Thus, it is easier for older adults in these settings to continuously engage in activities that are cognitively challenging.

- Line 226: This is another result that should be discussed carefully. The authors propose that smoking can “relax” older adults, and thus “protect” their cognitive function after the age of 70 years of age. The potential social implications of such a claim are significant, and I highly advise the authors to be critical of their results considering their study limitations (e.g., design, covariate definition, data collection). Moreover, it would be important to contrast your findings with other studies, which are lacking.

Answer: Thanks for your comments!  We have revised them as follows:

However, in this paper, the survival age of the regular smokers has not been considered. The survival age of the regular smokers may be lower than that of the less frequent smokers, therefore, some regular smokers lose their cognitive ability but their data are not recorded. The cognitive function of the existing smokers is higher than that of the infrequent smokers.

- The authors should address in detail other limitations related to the study design, the potential risk of interpretation bias, and the lack of a detailed instrument for data collection (which could explain some of the results found for covariates such as smoking).

Answer: Thanks for your comments!  We have revised them as follows:

Third, this study did not consider whether the surviving age of the frequent smokers will affect cognitive function. Future research is warranted to examine the impact of surviving age on the cognitive function of smoking older adults. The results of smoking around 68-70 should be considered their study limitations (e. g., study design, covariate definition, interpretation bias, data collection).

- The discussion section lacks a paragraph or two focused on the implications of your findings to clinical practice and social policy. What and whom do you expect to inform with such results?

Answer: Thanks for your comments!  We have revised them as follows:

 So, to prevent the decline of cognitive function of older adults, the government should build more playgrounds to facilitate the activities of older adults. People should also encourage older adults to participate in more social activities and persuade them to smoke and drink less.

Author Response

  1. The introduction needs to demonstrate the importance of the research topic, summarize recent research and knowledge gaps, and explain how this paper would make innovative and important contributions to the topic area.

Answer: Thanks for your comments! We have revised them and some paragraphs have been added. All revisions are marked in red in this revised paper. For example:

At present, more than 70 countries have entered an aging society, and China is one of them. According to the statistical bulletin of national economic and social development in China, by the end of 2019, there were 254 million older adults over the age of 60, accounting for 18.1% of Chinese total population. It is estimated that by 2033, the scale of older adults in China will reach 400 million (Jia et al., 2020). With the continuous increase of the global older adults, a series of public health and psychological problems, which are related to the cognitive function of older adults, seriously affect the physical and mental health and quality of life of older adults (Han et al., 2022). Some problems have gradually become prominent, such as Alzheimer’s disease. Alzheimer's disease not only severely affects the physical and mental health of older adults and the life quality in their later years, but also causes a heavy socio-economic burden on the family and society.

……

Although some studies have shown that there is a turning point of cognitive decline in older adults (Petersen et al., 2001; Zaninotto et al., 2018), this turning point has not been clearly explored, and it is still unknown which specific age is the real turning point. Therefore, it is necessary for us to understand this turning age and its influencing factors for cognitive decline of older adults, because it is very important and significant to recognize the turning point, which is conducive to be prepared in advance for cognitive decline of older adults.

……

Using PLGCM combined with a cohort-sequential design, this study intends to explore the staged changes in the cognitive function of Chinese older adults and obtain the turning point of the cognitive function of older adults. So that those people who serve the older adults can intervene the cognitive function of older adults before the turning point in the future older adults, which is conducive to providing older adults with a better life in their later years. On the one hand, this study aims to examine the development trajectory of cognitive function in older adults, and to understand the regularity of cognitive function in older adults. On the other hand, it includes different covariates to explore the influencing factors of cognitive function, in order to improve the recognition of older adults.

……

The research results will have certain theoretical and practical significance. On the one hand, it can help us understand the development model of the cognitive function of older adults, help clarify the work direction for older adults, and provide the theoretical basis for improving the older adults’ comfortable life. On the other hand, it explores the development of the cognitive function of older adults. Understanding the turning point of the trajectory is of great significance to the workers for older adults and older adults themselves. Before the turning point, protective interventions on cognitive function can protect the decline of cognitive function.

  1. Display characteristics and cognitive score by number of non-missing waves, since only 8.2% of the sample completed all 5 waves of survey.

Answer: Thanks for your comments! To avoid misunderstanding, we have deleted 8.2% of the expressions, and the expressions of missing values are shown in Table 2 for all 5 waves of survey.Revised as :

A total of 714 participants (55.9%) contributed data over all 5 age groups, and the missing data in each group is displayed in Table 2.

  1. Han, C., An, J., & Chan, P. (2022).Effects of cognitive ageing trajectories on multiple adverse outcomes among Chinese community-dwelling elderly population. BMC Geriatrics, 22(1), 692.

Answer: Thanks for your comments! We have read it carefully. This article has been referred to and this document has been added.

  1. This part fails to explain its main contributions and originality. A major revision in terms of innovation and contributions in the introduction and discussion will be important.

Answer: Thanks for your comments! We have revised the introduction and added a paragraph follows:

Using PLGCM combined with a cohort-sequential design, this study intends to explore the staged changes in the cognitive function of Chinese older adults and obtain the turning point of the cognitive function of older adults. So that those people who serve the older adults can intervene the cognitive function of older adults before the turning point in the future the elderly, which is conducive to providing older adults with a better life in their later years. On the one hand, this study aims to examine the development trajectory of cognitive function in the elderly, and to understand the regularity of cognitive function in older adults. On the other hand, it includes different covariates to explore the influencing factors of cognitive function, in order to improve the recognition of older adults.

We have revised the discussion and added a paragraph follows:

Using the PLGCM and cohort-sequential design that are new and creative, the turning point of cognitive decline in Chinese older adults is found, which has not been studied before. We find the two main risk factors and the two main protective factors influencing cognitive function in Chinese older adults, which is different from other studies.

  1. Additionally, how results from this paper are consistent/inconsistent with other studies about cognitive trajectories among Chinese older adults is important .

Answer: Thanks for your comments!  We have revised them and some new literature have been added. E.g.,

Franke, K., Ristow, M., & Gaser, C. (2014). Gender-specific impact of personal health parameters on individual brain aging in cognitively unimpaired elderly subjects. Frontiers in Aging Neuroscience, 6, 94.

Han, C., An, J., & Chan, P. (2022). Effects of cognitive ageing trajectories on multiple adverse outcomes among Chinese community-dwelling elderly population. BMC Geriatrics, 22(1), 692.

Jia, L., Du, Y., Chu, L., Zhang, Z., Li, F., Lyu, D., et al. (2020). Prevalence, risk factors, and management of dementia and mild cognitive impairment in adults aged 60 years or older in China: A cross-sectional study. Lancet Public Health, 5(12), e661–71.

Wu, C., Gao, L., Chen, S., & Dong, H. (2016). Care services for elderly people with dementia in rural China: a case study. World Health Organization. Bulletin of the World Health Organization, 94(3), 167–173.

An, R., & Liu, G. G. (2016). Cognitive impairment and mortality among the oldest-old Chinese. International Journal of Geriatric Psychiatry, 31(12), 1345–1353.

Jonkman, N.H., Panta, V.D., &Hoekstra, T., et al. (2018). Predicting Trajectories of Functional Decline in 60- to 70-Year-Old People. Gerontology, 64(3), 212–221.

Michael, J., Lyons, M.D., Grant, C.A., Reynolds, W.K., et al. (2017). A Longitudinal Twin Study of General Cognitive Ability Over Four Decades. Developmental Psychology, 53(6), 1170–1177.

Reviewer 3 Report

Thank you so much for the opportunity to provide some feedback and apologies for the extension to be able to evaluate this article.
 The manuscript is well organized and all the information is presented clearly. My recommendations for improving its content are:  1. Initially, you have to present other studies (longitudinal) to enhance the theoretical basis of your research project. 2) The aim of the study is clearly presented but not the hypothesis of your research idea. All the hypotheses must be supported by other studies, a fact that is not clear in the Introduction part. After the presentation of your study's main objective, I suggest clarifying your hypothesis.  3) You have to explain the statistical analysis before presenting your data. For example, why did you decide to conduct a Confirmatory Analysis? 4) You have to enhance your work with more references. some of the studies that you refer to are quite old. You must enhance your references. 

Author Response

Thank you so much for the opportunity to provide some feedback and apologies for the extension to be able to evaluate this article.

 The manuscript is well organized and all the information is presented clearly. My recommendations for improving its content are: 

  • Initially, you have to present other studies (longitudinal) to enhance the theoretical basis of your research project.

Answer: Thanks for your comments! We have revised the introduction and added some paragraphs follows:

Through longitudinal research, different scholars have studied different models and concluded that with the increase of the age, the cognitive function trajectory of older adults declines at different speeds. Jonkman et al. (2018) use polynomial regression model to find that the cognitive function of older adults showed a downward trend with the age. Michael et al. (2017) use the latent variable growth curve model to find that the older the age is, the faster the cognitive function of older adults declines. The results of Zaninotto et al. (2018) show that there is an inconsistent trend in the decline rate of the overall cognitive function of older adults around the age of 70-80.

Although some longitudinal studies have shown that there is a turning point of cognitive decline in older adults (Petersen et al., 2001; Zaninotto et al., 2018), this turning point has not been clearly explored, and it is still unknown which specific age is the real turning point. Therefore, it is necessary for us to understand this turning age and its influencing factors for cognitive decline of older adults, because it is very important and significant to recognize the turning point, which is conducive to be prepared in advance for cognitive decline of older adults.

  • The aim of the study is clearly presented but not the hypothesis of your research idea. All the hypotheses must be supported by other studies, a fact that is not clear in the Introduction part. After the presentation of your study's main objective, I suggest clarifying your hypothesis. 

Answer: Thanks for your comments! We have revised the introduction and added a paragraph follows:

The following hypotheses are proposed: Hypothesis 1: Investigate the general trajectory of the cognitive function development of older adults, and on this basis, examine the turning point of the cognitive function of older adults. Hypothesis 2: Including time-invariant covariates and time-varying covariates to investigate their impact on cognitive function of older adults.

  • You have to explain the statistical analysis before presenting your data. For example, why did you decide to conduct a Confirmatory Analysis?

Answer: Thanks for your comments! We have revised the introduction and added some paragraphs.

For example, in the Participants, we add one paragraph: To describe the basic information of data, the descriptive statistics are shown in Table 4.

For example, in the Participants, we add one paragraph: To indicate whether the data is randomly missing or not, we performed cross-tab chi-square tests on the selected data, and the chi-square value was not significant, χ2(df) =36.708, p = 0.436, indicating that the missing data was missing at random (Little, 1988).

For example, in the Results, we add one paragraph: In order to illustrate the characteristics of PLGCM, we analyzed the fitting indicators of the model. In PLGCM, the model fit indices indicated a good fit to the data (χ² = 32.06, df = 7, χ²/df= 4.6; CFI = 0.94; TLI = 0.90, RMSEA = 0.06, and SRMR = 0.04).

  • You have to enhance your work with more references. some of the studies that you refer to are quite old. You must enhance your references. 

Answer: Thanks for your comments!  We have revised them and some new literature have been added. E.g.,

Franke, K., Ristow, M., & Gaser, C. (2014). Gender-specific impact of personal health parameters on individual brain aging in cognitively unimpaired elderly subjects. Frontiers in Aging Neuroscience, 6, 94.

Han, C., An, J., & Chan, P. (2022). Effects of cognitive ageing trajectories on multiple adverse outcomes among Chinese community-dwelling elderly population. BMC Geriatrics, 22(1), 692.

Jia, L., Du, Y., Chu, L., Zhang, Z., Li, F., Lyu, D., et al. (2020). Prevalence, risk factors, and management of dementia and mild cognitive impairment in adults aged 60 years or older in China: A cross-sectional study. Lancet Public Health, 5(12), e661–71.

Wu, C., Gao, L., Chen, S., & Dong, H. (2016). Care services for elderly people with dementia in rural China: a case study. World Health Organization. Bulletin of the World Health Organization, 94(3), 167–173.

An, R., & Liu, G. G. (2016). Cognitive impairment and mortality among the oldest-old Chinese. International Journal of Geriatric Psychiatry, 31(12), 1345–1353.

Jonkman, N.H., Panta, V.D., &Hoekstra, T., et al. (2018). Predicting Trajectories of Functional Decline in 60- to 70-Year-Old People. Gerontology, 64(3), 212–221.

Michael, J., Lyons, M.D., Grant, C.A., Reynolds, W.K., et al. (2017). A Longitudinal Twin Study of General Cognitive Ability Over Four Decades. Developmental Psychology, 53(6), 1170–1177.

Round 2

Reviewer 1 Report

Dear Editor

I appreciate the opportunity to review the newest version of the manuscript titled “Turning Point of Cognitive Decline for Chinese Elderly from a Longitudinal Analysis: Protective Factors and Risky Factors”. The authors presented a retrospective secondary analysis that aimed to “explore the turning point of cognitive decline in Chinese elderly, and to explore the influencing factors including covariates”.

Overall, I believe the manuscript is well-structured and concise but is still requires major English revisions. The reference style will require revision from the authors to comply with MDPI’s format (including in the newly added textual segments).

[Title, abstract and keywords]

The authors should revise the sentence “At the same time, the government should also do a good job in promoting older adults to quit smoking and drinking”, since “good job” seems rather harsh. I would suggest something like “Given our findings, public health interventions centred on alcohol and tobacco cessation in older adults should be governmentally-endorsed”.

[Introduction]

The authors have rewritten quite extensively this section, introducing a more linear narrative that supports the study (congratulations). However, I would suggest:

·       The new text that was introduced on page 2 (line 82) to page 3 (line 139) be moved to the Method section;

·       The new text that was introduced on page 4 (line 145 until the end of the paragraph), is to be removed since it leans more toward a discussion of the implications of the study. Please finish your introduction section mentioning study objectives AND hypotheses (page 3, line 140).

[Method]

-        Table 2’s title is odd (Participants situation)? What do you mean? Sample size distribution (per cohort) during the study period?

-        Table 3, please remember that N means population, while n means sample.

[Results & Discussion]

Page 10, line 335: I still have a major issue with this paragraph,  given its potential ethical implications in public health. I would advise the authors to:

-        Revise the initial sentence to something like “Our findings show that after the transition stage, smoking does not significantly aggravate older adults’ cognition (…)”;

-        Remove the sentence that starts with “This may be because” entirely;

-        At the end, clearly write something like “While our findings are focused on the direct impact on older adults’ cognition, smoking in advanced ages is known to impact other health-related conditions such as ….” (while including recent references from other robust epidemiological studies).

On page 11, lines 381-383, the authors state that “the government should build more playgrounds to facilitate the activities of older adults”. While I understand the idea, the term “playground” is not the most suitable. I advise the authors to read the WHO initiative “Age-Friendly Cities” https://extranet.who.int/agefriendlyworld/age-friendly-cities-framework/  and revise the sentence.

Please, do not list your conclusions. 

Author Response

I appreciate the opportunity to review the newest version of the manuscript titled “Turning Point of Cognitive Decline for Chinese Elderly from a Longitudinal Analysis: Protective Factors and Risky Factors”. The authors presented a retrospective secondary analysis that aimed to “explore the turning point of cognitive decline in Chinese elderly, and to explore the influencing factors including covariates”.

Overall, I believe the manuscript is well-structured and concise but is still requires major English revisions. The reference style will require revision from the authors to comply with MDPI’s format (including in the newly added textual segments).

Answer: Thanks for your comments! We have revised it and the reference style  complis with MDPI’s format.

 [Title, abstract and keywords]

The authors should revise the sentence “At the same time, the government should also do a good job in promoting older adults to quit smoking and drinking”, since “good job” seems rather harsh. I would suggest something like “Given our findings, public health interventions centred on alcohol and tobacco cessation in older adults should be governmentally-endorsed”.

Answer: Thanks for your comments! We have revised it as :Given our findings, public health interventions centred on alcohol and tobacco cessation in older adults should be governmentally-endorsed.

All revisions are marked in blue in this revised paper.

 [Introduction]

The authors have rewritten quite extensively this section, introducing a more linear narrative that supports the study (congratulations). However, I would suggest:

  • The new text that was introduced on page 2 (line 82) to page 3 (line 139) be moved to the Method section;

Answer: Thanks for your comments! We have moved according your adivce. Thank you.

  • The new text that was introduced on page 4 (line 145 until the end of the paragraph), is to be removed since it leans more toward a discussion of the implications of the study. Please finish your introduction section mentioning study objectives AND hypotheses (page 3, line 140).

Answer: Thanks for your comments! We have removed it and add one paragraph as follows:

The purpose of this paper is as follows: (1)to find out the turning point of cognitive decline in Chinese older adults by cohort-sequential design through longitudinal research; (2)to find out the risk factors and protective factors that affect the decline of cognitive function in Chinese older adults through the Piecewise Latent Growth Curve Model (PLGCM).

[Method]

-        Table 2’s title is odd (Participants situation)? What do you mean? Sample size distribution (per cohort) during the study period?

Answer: Thanks for your comments! We have revised it as “Sample size distribution (per cohort) during the study period”.

-        Table 3, please remember that N means population, while n means sample.

Answer: Thanks for your comments! We have revised it.

 [Results & Discussion]

Page 10, line 335: I still have a major issue with this paragraph,  given its potential ethical implications in public health. I would advise the authors to:

-        Revise the initial sentence to something like “Our findings show that after the transition stage, smoking does not significantly aggravate older adults’ cognition (…)”;

Answer: Thanks for your comments! We have revised it as follows:itive function of older adults will decline (Li et al., 2017)[32].

Our findings show that before and after the turning point, smoking is an influencing factor, which is consistent with some studies (Liu et al., 2002)[33]. However, in this paper, the survival age of the regular smokers has not been considered. The survival age of the regular smokers may be lower than that of the less frequent smokers, therefore, some regular smokers lose their cognitive ability but their data are not recorded. The cognitive function of the existing smokers is higher than that of the infrequent smokers. While our findings are focused on the direct impact on older adults’ cognition, the smoking in advanced ages is known to impact other health-related conditions such as lung disease, asthma, stroke, etc. (Aune et al., 2016; Larsson et al., 2020; Stefanidou et al., 2022)[34-36].

-        Remove the sentence that starts with “This may be because” entirely;

Answer: Thanks for your comments! We have remove it.

-        At the end, clearly write something like “While our findings are focused on the direct impact on older adults’ cognition, smoking in advanced ages is known to impact other health-related conditions such as ….” (while including recent references from other robust epidemiological studies).

Answer: Thanks for your comments! We have added it as follows:

While our findings are focused on the direct impact on older adults’ cognition, the smoking in advanced ages is known to impact other health-related conditions such as lung disease, asthma, stroke, etc. (Aune et al., 2016; Larsson et al., 2020; Stefanidou et al., 2022)[34-36].

On page 11, lines 381-383, the authors state that “the government should build more playgrounds to facilitate the activities of older adults”. While I understand the idea, the term “playground” is not the most suitable. I advise the authors to read the WHO initiative “Age-Friendly Cities” https://extranet.who.int/agefriendlyworld/age-friendly-cities-framework/  and revise the sentence.

Answer: Thanks for your comments! After we read the WHO initiative “Age-Friendly Cities”,we revised it as follows:

the respect form the government should be reflected in the accessibility of public buildings and spaces and in the range of opportunities that the city offers to older adults for social participation, entertainment, volunteering etc. Given our findings, public health interventions centred on alcohol and tobacco cessation in older adults should be governmentally-endorsed.

Reviewer 2 Report

I appreciate that the authors improved the manuscript significantly in terms of its originality, contributions, and implications. The introduction and discussion need some additional improvement. It is particularly important to distinguish between implications from findings about levels and about trends in cognitive function. 

1. Line 52: Cognitive decline is not necessarily a disease. Rather, it can be an early manifestation of dementia. 

2. Line 56: Preventing or delaying the early onset of dementia will reduce its burdens. 

3. Line 82: clarify gaps in knowledge about the turning point of cognitive decline and gaps in knowledge about cognitive decline using longitudinal studies separately to improve the clarity of the text. 

4. Line 86: Practice effects in longitudinal studies may also be a disadvantage. 

5. Line 130: Clarify whether this study is the first one to combine PLGCM and cohort-sequential design to study the trajectory of cognitive function and associated risk/protective factors. 

6. Line 279: Clarify how risk/protective factors affected levels or trends in cognitive function. Clarify the outcome - whether it refers to levels or trends of cognitive function. 

7. Line 290: In the discussion section, highlight the contribution of this study in terms of the development trajectory of the cognitive function of Chinese older adults and factors influencing the development trajectory of cognitive function respectively.  Explain explicitly what additional knowledge compared to previous literature this study provided. Separately clarify implications from findings about factors associated with levels and associated with trends in cognitive function. 

Author Response

I appreciate that the authors improved the manuscript significantly in terms of its originality, contributions, and implications. The introduction and discussion need some additional improvement. It is particularly important to distinguish between implications from findings about levels and about trends in cognitive function. 

  1. Line 52: Cognitive decline is not necessarily a disease. Rather, it can be an early manifestation of dementia. 

Answer: Thanks for your comments! We've added it to the text. All revisions are marked in blue in this revised paper.

  1. Line 56: Preventing or delaying the early onset of dementia will reduce its burdens. 

Answer: Thanks for your comments! We've added it to the text. All revisions are marked in blue in this revised paper.

  1. Line 82: clarify gaps in knowledge about the turning point of cognitive decline and gaps in knowledge about cognitive decline using longitudinal studies separately to improve the clarity of the text.

Answer: Thanks for your comments! To illustrate this, we add a paragraph in Introdcution as follows:

However, most of the previous studies were based on cross-sectional studies (Li & Chen, 2003; Drag & Bieliauskas, 2010; Wu et al., 2016)[5-7]. Even longitudinal studies rarely discussed the specific time turning point of cognitive decline in the elderly, and rarely analyzed its influencing factors (Han et al. 2022; Hou et al., 2018)[2,11], which is insufficient. Therefore, it is meaningful to study the turning point of cognitive decline in Chinese older adults and its main influencing factors, which is conducive to preparing Chinese older adultst and their service personnel in advance.

  1. Line 86: Practice effects in longitudinal studies may also be a disadvantage.

Answer: Thanks for your comments! We've added it to the text. All revisions are marked in blue in this revised paper.

  1. Line 130: Clarify whether this study is the first one to combine PLGCM and cohort-sequential design to study the trajectory of cognitive function and associated risk/protective factors. 

Answer: Thanks for your comments! Yes, it is first time. Using the PLGCM and cohort-sequential design that are new and creative, the turning point of cognitive decline in Chinese older adults is found, which has not been studied before.

  1. Line 279: Clarify how risk/protective factors affected levels or trends in cognitive function. Clarify the outcome - whether it refers to levels or trends of cognitive function. 

Answer: Thanks for your comments! We add some sentence to illustrate it as follows:

With the aging, the cognitive function level of older adults shows a downward trend and decreases rapidly after the age of 68-70. Education level significantly predicted (β = 0.032, SE = 0.016, p < 0.01) the change rate of cognitive function in older adults after the age of 68-70 (see in table 6). High level of education(β = 0.032, SE=0.016) has a significant positive predictive effect on the change rate of cognitive function level after the turning point. That is, the cognitive function level of Chinese older adults with high education level declines more rapidly.

  1. Line 290: In the discussion section, highlight the contribution of this study in terms of the development trajectory of the cognitive function of Chinese older adults and factors influencing the development trajectory of cognitive function respectively.  Explain explicitly what additional knowledge compared to previous literature this study provided. Separately clarify implications from findings about factors associated with levels and associated with trends in cognitive function. 

Answer: Thanks for your comments! To illustrate this, we add two paragraphs in Introdcution as follows:

Compared with previous studies (Han et al. 2022; Hou et al., 2018; Petersen et al., 2001; Zaninotto et al., 2018)[2,4,10,11], the contribution of this study is mainly shown in the following two points: (1) the specific turning point of cognitive decline of the elderly was found, namely 68-70. In fact, understanding the turning point of the trajectory is of great significance to the workers for older adults and older adults themselves; (2)Four main factors influencing the turning point of cognitive decline in the elderly were explored. Before the turning point, protective interventions on cognitive function can protect the decline of cognitive function.

This research explores the development of the cognitive function of older adults and can help us understand the development model of the cognitive function of older adults, and help clarify the work direction for older adults, and provide the theoretical basis for improving the older adults’ comfortable life.
